

# Spatial and Temporal Pattern of Drought Hazard under Different RCP Scenarios for China in the 21[st] century

Tao. Pan [1], Jie. Chen [1,2], Yujie. Liu [1]

[1] Institute of Geographic Sciences and Natural Resources Research, Chinese Academy of Sciences (CAS), Beijing, 100101,
PR China
[2] University of Chinese Academy of Sciences (UCAS), Beijing, 100049, PR China

*Correspondence to*: Yujie. Liu (liuyujie@igsnrr.ac.cn)

**Abstract.** Drought is one of the most common natural disasters with significant negative impacts on socio-economic development and the natural environment. Evaluating this hazard is essential for risk assessment and management. In this study, historical climate data from 1961 to 2010 were used to determine the appropriate timescales for applying the standardized precipitation evapotranspiration index (SPEI) to evaluate drought. Simulated representative concentration pathway (RCP) 4.5 and 8.5 climate data for 2011–2099 in China was obtained from the Inter-Sectoral Impact Model
Intercomparison Project (ISI-MIP). Then, the Mann–Kendall (M-K) test was used to assess the significance of drought trends. The frequency of three drought grades and duration were chosen to reflect the spatial and temporal variation in drought hazards in three time periods, 2011-2040, 2041-2070 and 2071-2099. Results indicate that 2071–2099 will face the most severe droughts, with the highest frequency (32.54% for RCP4.5 and 32.55% for RCP8.5) and longest duration (3.93 months for RCP4.5 and 4.12 months for RCP8.5), followed by 2041–2070 and 2011–2040. In terms of spatial distribution, drought hazards
in north China (Medium temperate zone, Warm temperate zone and Plateau temperate zone) will be greater than in south China, especially the non-monsoon region. Comparing the two scenarios, drought hazards from RCP8.5 are higher than that from RCP4.5, i.e., higher frequency (32.35% for RCP8.5 and 32.26% for RCP4.5), longer duration of drought (3.84 months for RCP8.5 and 3.76 months for RCP4.5) and more significant drying trends. These results provide a reference for adapting to extreme climate change and the prevention and reduction of disaster risks.

## 1 Introduction

Drought is one of the most common but devastating natural disasters in the world, and has significant impacts on socio-economic development and natural ecosystems. Due to a typical continental monsoon climate, droughts have large impacts on China, with high frequency, wide distribution, long duration, and clear seasonal characteristics (Qin et al., 2015). In 2015, the affected people of droughts are over 10 million and the economic losses caused by drought are more than 7 billion dollars in
China (CMA, 2016). Furthermore, drought risks are likely to increase as the increasing frequency and intensity of extreme climate events with global warming (Field et al., 2012; Stocker et al., 2013). Therefore, risk assessment is urgently needed in China to form effective and timely adaption and mitigation strategies. Risk is often represented as the probability of occurrence




of hazardous events or trends multiplied by the impacts of their occurrence, i.e., it is the interaction of hazard, exposure, and vulnerability; among these, hazard usually refers to climate-related physical events, trends, or their physical impacts (Field et al., 2014). Evaluating hazards is the most important aspect of risk assessment and the basis for further exposure and vulnerability studies (Hao et al., 2012).

Recently, extensive research has been conducted on variations in drought hazard in the context of global climate change (Cook et al. 2014; Nam et al. 2015; Quiroga and Suárez, 2016; Panda 2016). Gutzler and Robbins (2011) focused on changes in drought in semi-arid areas in the western United States based on the Palmer Drought Severity Index (PDSI), and found that drought would further expand over the next 100 years. Kim et al. (2016) analyzed the drought situation in Korea in RCP8.5 based on the standardized precipitation evapotranspiration index (SPEI), and showed that drought would become more serious

due to climate change. Similar research in China has primarily focused on regional studies, with fewer studies at the national level (Weng et al., 2015; Xing-Guo and Lin, 2015). In addition, most previous work in China assessed drought hazard indirectly by calculating the frequency of drought, while duration was seldom considered because it is difficult to define when a drought starts and ends. Finally, previous research has concentrated on historical droughts (Wang et al., 2011; Yu et al., 2014), and studies projecting future drought in the context of climate change, especially in representative concentration pathway (RCP)

scenarios, remain limited. RCP scenarios represent pathways based on simulated influences of land use and emissions of aerosols and greenhouse gases (GHG), the four RCPs together span the range to year 2100 with radiative forcing values from 2.6 to 8.5W/m$^2$ found in open literature. The highest pathway is RCP8.5 without applying any mitigation policy to GHG emissions. The lower scenarios (RCP2.6, RCP4.5 and RCP6.0), do adopt some mitigation measures to control the GHG emission (Vuuren et al., 2011). Because there has been insufficient hazard assessment for China, it is essential to conduct

research on the spatial and temporal patterns of drought hazard for China using different indices for the future under RCP scenarios.

     Drought indexes include the PDSI, SPEI, Standardized Precipitation Index (SPI), Integrated Surface Drought Index (ISDI), and Vegetation Health Index (VHI) (Palmer, 1965; Vicente-Serrano et al., 2010b; McKee et al., 1993; Wu et al., 2013; Rhee et al., 2010). SPEI, SPI, and PDSI are more widely used. SPI, with multi-scale characteristics, can reflect different types of

drought, but does not consider temperature, humidity, and other factors that affect drought (Hayes et al., 1999). PDSI accounts for both precipitation and evapotranspiration factors, which can reflect the effect of warming on drought; however, PDSI does not have multi-scale characteristics (Dubrovsky et al., 2009). Combining the multi-scale characteristics of the SPI with PDSI, which is sensitive to warming, Vicente-Serrano et al. (2010a) proposed the SPEI, which is easily calculated and expanded to large scales (Wang et al., 2015). With these advantages, SPEI has been widely applied to drought research to evaluate future

hazards due to climate change (Vicente-Serrano et al., 2011; Deo and Şahin, 2015; Touma et al., 2015).

     The primary objective of this study was to assess the spatial and temporal variations in drought hazard across China under different RCP scenarios. First, we determined the most suitable timescale for analyzing the SPEI based on measured data. Second, the Mann–Kendall (M-K) test (Mann, 1945; Kendall, 1955) was used to evaluate the significance of wetting/drying



trend based on simulated data. Third, frequency of drought at three levels of intensity and drought duration were chosen as indices to project the spatial and temporal variation of drought hazard.

## 2 Materials and methodology

### 2.1 Eco-geographical regionalization for China

To reveal regional differences in drought hazard, China was divided into different eco-geographic regions for analysis. Eleven temperature zones were defined according to differences in surface topographic features and geomorphic structure, combinations of temperature and water conditions, and zonal vegetation and soil type (Fig. 1) (Zheng, 2008). The equatorial tropical zone was not included in this data.

### 2.2 Data processing

Observation data from 542 meteorological stations (Fig. 1) for 1961–2010 were obtained from the China Meteorological Administration. Monthly meteorological data included precipitation, average maximum temperature, average minimum temperature, average wind speed, average relative humidity, sunshine hours, and solar radiation. Simulated scenario data were obtained from Inter-Sectoral Impact Model Intercomparison Project (ISI-MIP) (Warszawski et al., 2014), which contains five sets of GCM results (GFDL-ESM2M, HadGEM2-ES, IPSL-CM5A-LR, MIROC-ESM-CHEM, NorESM1-M) for the RCP

scenarios (Tab.S1). The output data have been converted to a degree latitude / longitude 0.5° x 0.5° resolution by spatial downscaling and have been offset corrected. We synthesized the simulation results for these five GCMs because the combined effect of multiple models is superior to that of single model (Zhou and Yu, 2006). RCP4.5 and RCP8.5 were selected for analysis in this study. The scenario data were divided into three time periods, 2011–2040, 2041–2070, and 2071–2099.

### 2.3 Calculating the SPEI and M-K trend test

SPEI uses the log-logistic probability distribution function to reflect the change in water deficit, and obtains the drought index value by standardizing (Vicente-Serrano et al., 2010a). Differences between precipitation (P) and potential evapotranspiration ($ET_0$), which reflect the water surplus or deficit in a region, were calculated to deduce the SPEI by using:

$$D = P - ET_0$$

The Thornthwaite (1948) equation for calculating $ET_0$ only takes temperature into account, ignoring the effects of wind

speed, relative humidity, solar radiation, and other dynamic factors on drought. Therefore, we used the Penman-Monteith equation recommended by FAO (1998) in this study, which accounts for both thermal and dynamic factors; therefore, the results are more consistent with actual reference crop evapotranspiration. The radiation coefficient used was based on the radiation calibration results in China by Yin et al. (2008):



$$ET_0 = \frac{0.408\Delta(R_n - G) + \gamma \frac{900}{T + 273} u_2(e_s - e_a)}{\Delta + \gamma(1 + 0.34)u_2}$$

Here, $ET_0$ is the potential evapotranspiration; $Rn$ is the net radiation; $G$ is the soil heat flux density; $T$ is the surface mean daily air temperature; $u_2$ is the wind speed at 2 m height above the ground; $e_s$ is the saturation vapor pressure; and $e_a$ is the actual vapor pressure. The SPEI was calculated by the R-SPEI-package (https://CRAN.R-project.org/package=SPEI). The input data is monthly time series of D, where the set parameter are kernel = 'rectangular', distribution = 'log-Logistic', and fit = 'ub-pwm'. SPEI categories for drought grade and its probability as well as the definition are shown in Tab. 1 (Liu and Jiang, 2015).

The nonparametric M-K test was used to calculate trends in SPEI, which is commonly used in meteorology and hydrology (Mann, 1945; Kendall, 1955). A positive value of Z indicates an upward trend, while a negative value of Z indicates a downward trend. |Z|> 1.96 indicates a significant upward/downward trend at 0.05 significance level, and |Z|> 2.576 indicates an extremely significant upward/downward trend at 0.01 significance level (Zarch et al., 2015). In this study, we chose a significance level of 0.05 for analysis. The M-K test was calculated by the R-Kendall-package (https://cran.r-project.org/web/packages/Kendall/index.html).

## 2.4 Drought hazard indices

Common indices for drought hazard assessment include frequency, intensity, and duration. We quantified drought duration and frequency of the three drought intensities, mild, moderate, and extreme, to reveal the spatial and temporal variation of drought hazards across China for the two RCP scenarios.

### 2.4.1 Drought frequency

The drought frequency is calculated as:

$$Pi = \frac{n_i}{N} * 100\%$$

Where, $N$ is the total number of months to calculate, $n_i$ is the number of months that drought has taken place in the $i$th station. In this study, the frequency of each grid for mild, moderate, and extreme droughts in RCP4.5 and RCP8.5 scenarios were calculated.

### 2.4.2 Drought duration

The definition of drought duration used is the number of months in which moderate and extreme drought occurred during the study period divided by the number of drought events (Rhee and Cho, 2015). A drought event is determined by the start and termination time of drought, that is, from the month with SPEI < -0.5 to the end of the month with SPEI ≥ -0.5. Large values indicate the existence of persistent drought events with longer lengths, while small values indicate intermittent occurrences of short drought events (Spinoni et al., 2014).





## 3 Results

### 3.1 Determining SPEI over different timescales

SPEI was calculated for 1-, 3-, 6-, 12-, and 24-month timescales for 1961–2010 (Fig. S1). After comparing the characteristics of different SPEI timescales, we found that shorter timescales (SPEI-1, SPEI-3, and SPEI-6) reflect details of drought processes,

and are sensitive to extreme events. The longer timescale (SPEI-24) reflects long-term drought trends, but some drought events may be smoothed and missing. SPEI-12 also reflects long-term trends, while maintaining inter-annual drought changes and is not too sensitive to extreme events. Therefore, SPEI-12 was chosen for the M-K test, and to calculate the frequencies and durations of droughts. In general, drought frequency in China has been high for the past 50 years, and both frequency and duration have been increasing over time.

### 3.2 SPEI trends in different RCP scenarios

Time series for SPEI-12 were calculated for both RCP4.5 and RCP8.5, and SPEI trends for each temperature zone (excluding the equatorial tropical zone) and national average were obtained.

In RCP4.5 (Fig. 2) and RCP8.5 (Fig. 3), national average SPEI value changes from positive to negative around 2050, indicating a turning point in the wet-drought pattern for the entire country. However, there are some differences between

temperature zones and scenarios. There is a persistent drying trend in MT, WT, PS, and PT after the 2050s, while other regions show alternating drying and wetting trends. In RCP4.5, drought occurs primarily as mild, with a small number of moderate droughts and nearly no extreme droughts; the overall drought situation in China is not serious in this scenario. In contrast, in RCP8.5, the frequency of moderate drought increases significantly. The minimum value of SPEI in CT, MaT, and MiT is about -2. Considering the value reflects average level of drought in each temperate zone, in these zones with SPEI below -2,

extreme droughts are likely to occur. The degree of drought in RCP8.5 is more serious than in RCP4.5, and China will face a more severe drought situation if this scenario is more accurate for projecting future trends.

The Z values derived from application of M-K test corresponding to each grid in RCP4.5 and RCP8.5 were mapped to show the spatial distribution of SPEI trends (Fig. 4). In RCP4.5, northwest China shows downward (drying) trends: northern PS and PT show a downward trend, and WT and western MT present significant downward trends at the 0.05 significance level. CT,

PS, and southern China shows non-significant upward (wetting) trends. The drying trend from drought in RCP8.5 is more significant. The extent of the wetting trend in southern China is reduced, while drying trends in the WT and Qinghai-Tibet Plateau increase, and PT in the northern Qinghai-Tibet Plateau shows a significant drying trend.



### 3.3 Spatial and temporal patterns in drought hazard for different scenarios

#### 3.3.1 Spatial and temporal variation in drought frequency

The frequencies of mild, moderate, and extreme droughts in RCP4.5 and RCP8.5 were quantified for 2011–2040, 2041–2070, and 2071–2099, and mapped to show the spatial and temporal variations in drought frequency (Fig. S2, Fig. 5, and Fig. 6).

In RCP4.5 (Fig. S2a) and RCP8.5 (Fig. S2b), the national average frequency of mild drought is close to 15% (Tab. S2) in three time periods. As for spatial distribution, areas with high frequency of mild droughts are scattered, with no clear spatial trends. For the three periods, the frequency for 2041–2070 is the highest with the widest coverage area in both two scenarios, the frequency for 2011-2040 is the least in RCP4.5 while 2071-2099 is the least in RCP8.5 (Tab. S2).

    The spatial and temporal patterns for moderate drought frequency varies across periods, regions, and scenarios (Fig 5).
Temporally, moderate drought frequency first decreases and then tends to increase in the two scenarios. In detail, drought frequency in 2071–2099 is the highest, followed by 2011-2040, and 2041-2070(Tab. S2). Regionally, moderate drought occurs more frequently in northern versus southern China, but the frequency in southern China clearly increases in different periods. For example, in RCP4.5 (Fig. 5a), the moderate drought occurs mainly in northern China (MT, WT) for 2011–2040 and 2041-2070. However, the frequency particularly increases in south China (NS, MS, SS) for 2071-2099. In RCP8.5 (Fig. 5b), the
frequency of moderate drought in northwest inland and eastern coastal China is quite high for 2011–2040; parts of WT and PT reach frequencies > 20%. For 2041–2070, the frequency of moderate drought in northern China declines, while in south China moderate drought frequency slightly increases. For 2071-2099, areas with drought frequency > 20% significantly expand in MT and WT. Comparing the two scenarios, the frequency of moderate drought in RCP8.5 is higher than in RCP4.5 during the same time period (Tab. S2).

While extreme drought frequency is significantly less than for mild and moderate drought, there are clear spatial and temporal variations in both scenarios. The frequency of extreme drought in 2041–2070 is the highest, while it is the lowest in 2071–2099 in both two scenarios (Tab. S2). Spatially, extreme drought in RCP4.5 occurs primarily in southern China, while the frequency in northern China is higher in RCP8.5. More specifically, in RCP4.5 (Fig. 6a), extreme drought occurs mainly in NS, MS, SS, and the southern Qinghai-Tibet Plateau in 2011–2040. For 2041–2070, MT and WT have expanded areas of
extreme drought compared to the earlier period. Then, drought frequency clearly decreases in 2071–2099. In RCP8.5 (Fig. 6b), extreme drought occurs mainly in MT and eastern inland China in 2011–2040. In 2041–2070, the frequency of extreme drought in WT increases, while in southern China it decreases. Generally, extreme frequency in China during 2071-2099 decreases, except in the Qinghai-Tibet Plateau.

#### 3.3.2 Spatial and temporal variation in drought duration

Drought duration in RCP4.5 and RCP8.5 were calculated for the same three periods, and mapped to show the spatial and temporal variation (Fig. 7) across periods, regions, and scenarios.



Temporally, drought duration shows an increasing trend for two scenarios and all time periods. That is, drought duration in 2071–2099 is the longest, followed by 2041–2070, and the shortest is in 2011–2040 (Tab. S3). The durations tend to last between two and six months in most areas, the percentage of area is more than 90% in each time period and scenario. Spatially, drought durations are longer in northern China than in the south, with particularly severe droughts in northwest (WT) China.

Specifically, in RCP4.5 (Fig. 7a), drought duration in northern China is longer than in southern China in 2011–2040, and durations exceed six months in northwest China, including MT, WT, and PT. During 2041-2070, drought duration in the Qinghai-Tibet Plateau increases significantly compared to the 2011-2040. And the duration increase in 2071-2099 especially in western MS and SS.

In RCP8.5 (Fig. 7b), drought duration in MS is clearly longer than in RCP4.5 for 2011–2040, but the national drought duration is relatively shorter (Tab.S3). For 2041–2070, the duration increases significantly in CT, MT, MS, and SS. Northwest China faces the longest duration, and the duration in MT, MS, and the Qinghai-Tibet Plateau exceeds six months, even reaching more than eight months in some locations. For 2071-2099, the duration in northern China and the Qinghai-Tibet Plateau increases, while duration in southern China is reduced. Comparing the same time period in the two scenarios, for 2011-2040, drought durations in RCP8.5 (3.68) are slightly shorter than that in RCP4.5(3.84) while for 2041-2070 and 2071-2099, drought durations in RCP8.5(4.11, 4.12 respectively) are longer than that in RCP4.5(3.87, 3.93 respectively).

## 4 Discussion

Based on simulated scenario data from ISI-MIP, the spatial and temporal patterns of drought hazard was evaluated for China in RCP4.5 and 8.5 in 21$^{st}$ century. The outcome of this study indicates a continuing tendency for dry conditions in China is projected under RCP4.5 and RCP8.5 scenarios, but with large spatial and temporal variations over different time periods and regions. Among the three time periods, drought hazards in 2071–2099 are the most severe, with the highest frequency of drought (32.54% for RCP4.5 and 32.55% for RCP8.5), and longest duration (3.93 for RCP4.5 and 4.12 for RCP8.5), followed by 2041–2070 and 2011–2040. One explanation is that temperatures is gradually rising as time goes by, leading to clear increases in evapotranspiration and frequent and long duration drought events. Among all temperature zones, higher frequency and longer duration drought events are projected for MT, WT, and PT, all in northwest China. Increases in droughts are generally due to increases in evapotranspiration and decreases in precipitation. Therefore, with global warming, increases in evapotranspiration are projected for all regions of China. However, northwest China is remote from the ocean, which leads to losses in water vapor during transport as part of the typical continental monsoon climate. Furthermore, in the future, climate change will result in constant water storages and clear decreases in precipitation. Together, the increase in evapotranspiration under global warming coupled with typical continental monsoon climate will have important roles in exacerbating drought in these areas. This general result is in agreement with those from Huang et al. (2017). However, we find different spatial patterns in future droughts across China from other studies. For example, Li and Pan (2016) showed that the areas with the highest drought hazard in China are Sichuan, Guizhou, Yunnan, Gansu and Ningxia in RCP4.5 and 8.5, mainly located in MS and SS,



southern inland China. These differences may be attributed to different simulated scenario data and drought indexes used in the studies, which leads to uncertainties in estimating drought hazard. Comparing the two scenarios (Fig. 4, Fig. S4) for 2011-2099, drought hazards in RCP8.5 are higher than in RCP4.5, with higher frequency (32.35% for RCP8.5 and 32.26% for RCP4.5), longer duration (3.84 months for RCP8.5 and 3.76 months for RCP4.5), and more significantly downward (drying) trends. RCP8.5 is a very high baseline emission scenario; therefore, higher temperature and potential evapotranspiration eventually cause higher drought hazards. Considering simulated population and GDP data in shared socioeconomic pathways (SSPs) (O'Neill et al., 2014) correspond to RCP4.5 and RCP8.5, drought risk in RCP8.5 will be much more serious in RCP4.5, a result consistent with the findings of Zhang and Zhang (2016).

There are some uncertainties in estimating drought hazards under climate change. The main sources include GHG emission scenarios (Maurer, 2007), GCMs (Kirono et al., 2011), selection of drought index (Burke and Brown, 2008), calculating of potential evapotranspiration. The widely used IPCC scenarios only propose several possible scenarios, and some factors remain uncertain, such as population growth, the scale of economic development, energy structure changes, and progress in science and technology, which directly affect GHG emissions in the future. Furthermore, although many models have been developed to simulate global climate change, and studies have made great progress in developing computer technology and physical research, the simulated results from different models have differences. As for the chosen drought index, each index considers different factors, which leads to varying results. These factors include differing potential evapotranspiration models used to calculate the SPEI. Therefore, future studies can evaluate different drought indexes for calculation and analysis based on more advanced and higher resolution GCMs and RCMs (regional climate models), determine the importance of sources of uncertainty, and generate more accurate and reasonable drought assessment results.

**5 Conclusion**

This study used the SPEI to assess the spatial and temporal characteristics of future droughts in China. Drought frequency, duration, and trends were evaluated in three time periods under RCP4.5 and 8.5 scenarios. The conclusions can be summarized as follows. (1) Among three time periods, 2071–2099 will have the highest frequency and longest duration of drought, compared to 2041–2070 and 2011–2040. (2) Drought frequency and duration in northern China, including MT, WT, and PT, will be greater than in southern China, especially in the non-monsoon region. (3) Drought hazards will be greater under conditions modeled in RCP8.5 than those in RCP4.5, with higher frequency and longer duration droughts and more significant drying trends. The conclusions from this study provide a scientific basis for future drought vulnerability and risk assessments, which can guide policies for disaster prevention and mitigation.



**Acknowledgements**

This study was supported by the National Natural Science Foundation of China [Grant No. 41671107 and 41530749] and the Youth Innovation Promotion Association, CAS [Grant No. 2016049]. We also thank the ISI-MIP and China Meteorological Administration for providing data support.

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

20

25

30





**Tables**

**Table 1. Drought grade categories and probability in the SPEI and its definition.**

| SPEI | Categories | Probability | Definition |
|---|---|---|---|
| >-0.5 | Normal and wetness | 69.15% | Precipitation is normal or more than normal, surface is wet and there is no drought |
| -1.0~-0.5 | Mild drought | 14.98% | Precipitation is less than normal, surface air is dry, and soil moisture is insufficient |
| -2.0~-1.0 | Moderate drought | 13.59% | Precipitation continued to be less than normal, surface is dry, soil moisture is insufficient, which has a certain impact on crops and ecological environment |
| ⩽-2.0 | Extremely drought | 2.28% | Soil moisture is seriously deficient for a long time, which has a serious impact on crops, ecological environment, industrial production as well as drinking water for people and animals |

**Figures**

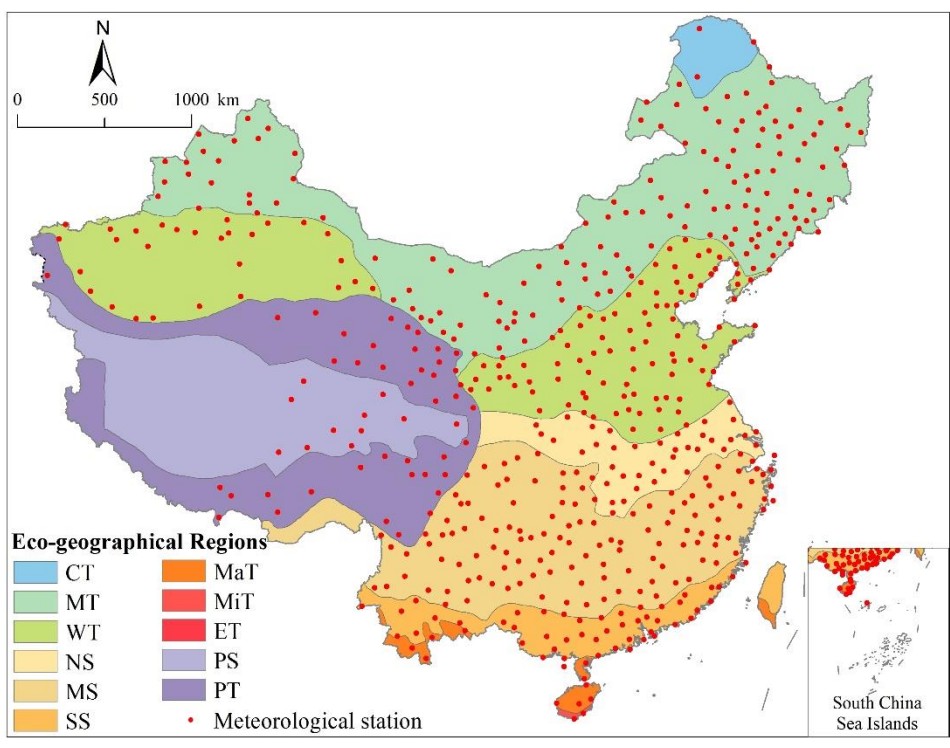

5  **Figure 1: Eco-geographic regional system for China and distribution of the 542 meteorological stations (CT-Cold temperate zone, MT-Medium temperate zone, WT-Warm temperate zone, NS-North subtropical zone, MS-Middle subtropical zone, SS-South subtropical zone, MaT-Margin tropical zone, MiT-Middle tropical zone, ET-Equatorial tropical zone, PS-Plateau sub-cold zone, PT-Plateau temperate zone).**


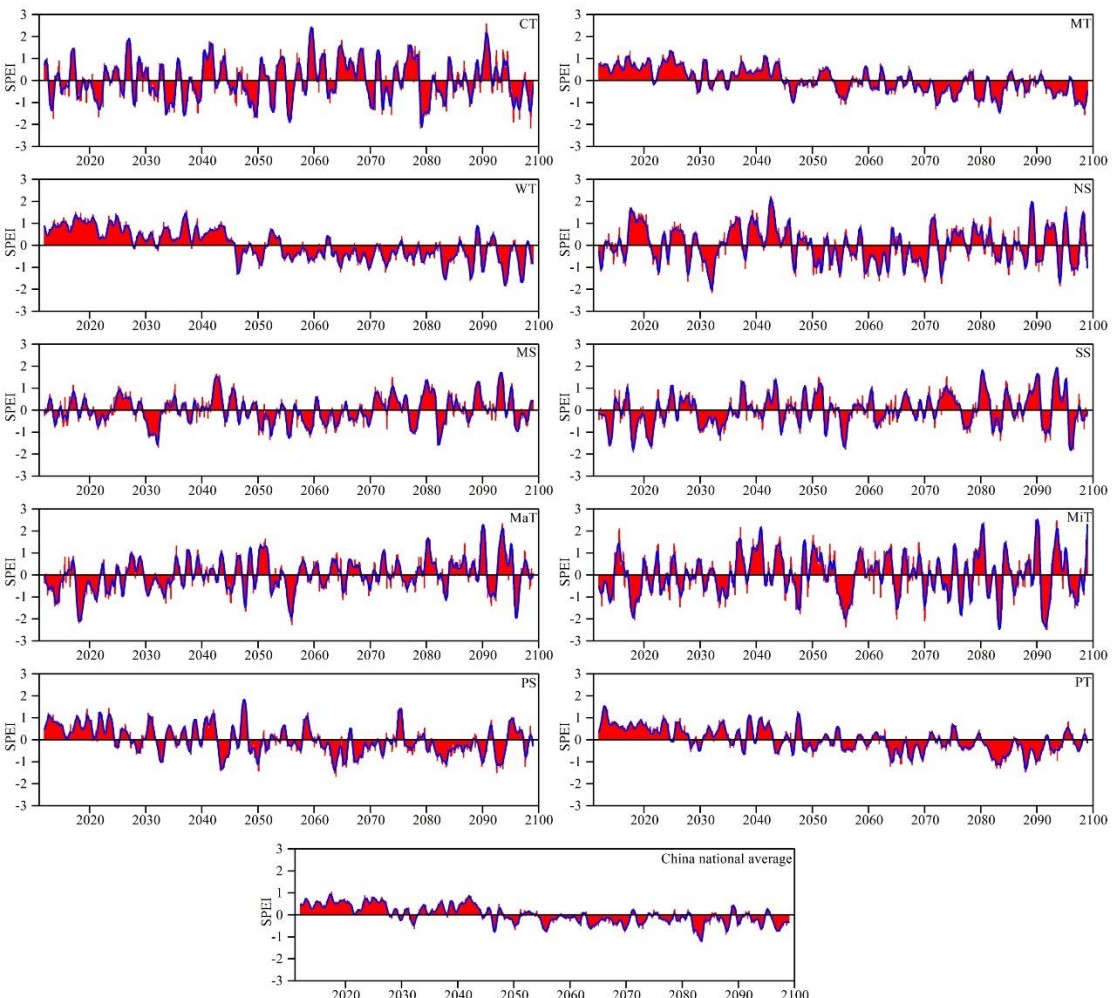

**Figure 2: SPEI trends in China for 2011–2099 in RCP4.5 scenario (CT, MT, WT, NS, MS, SS, MaT, MiT, PS, PT, and the national average).**





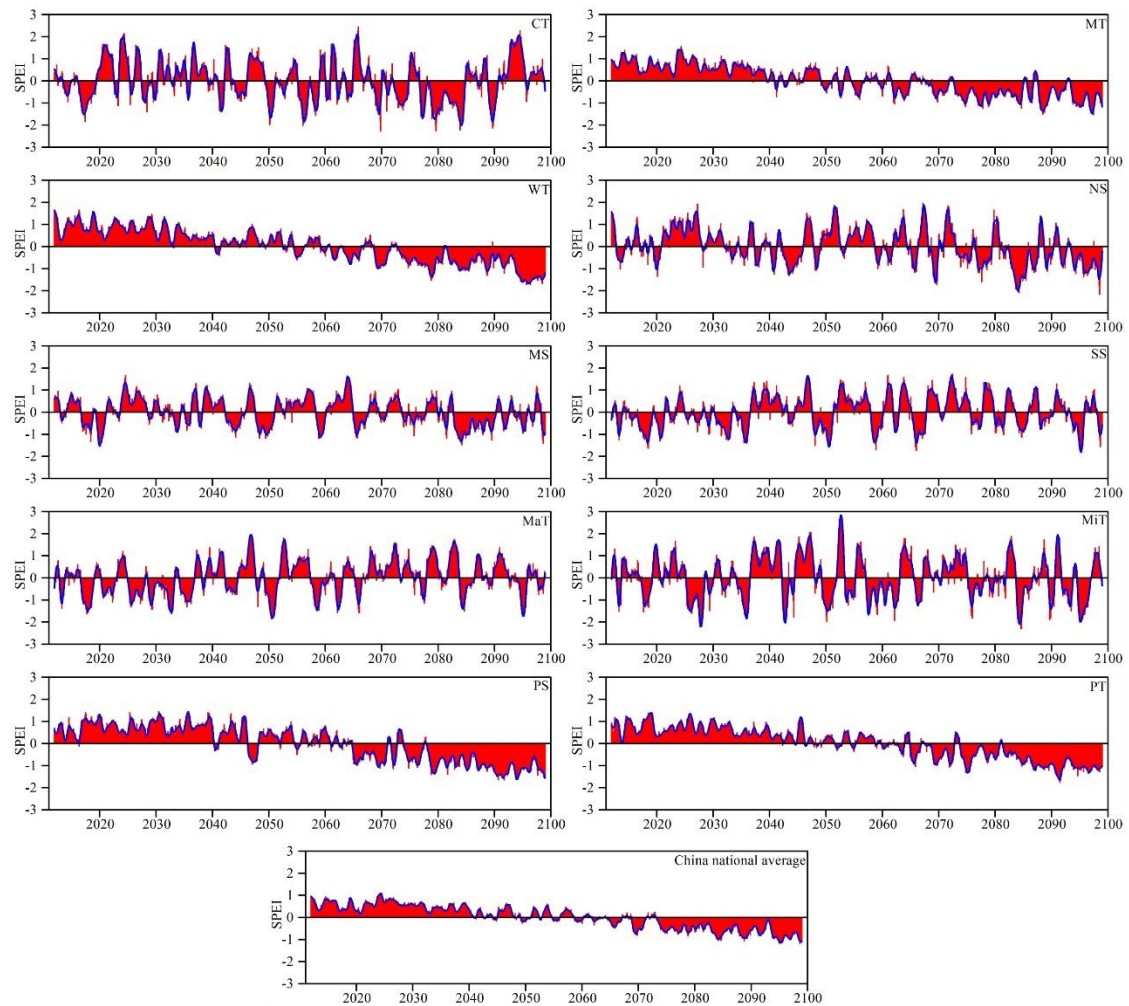

**Figure 3: SPEI trends in China for 2011–2099 in RC8.5 scenario (CT, MT, WT, NS, MS, SS, MaT, MiT, PS, PT, and the national average).**

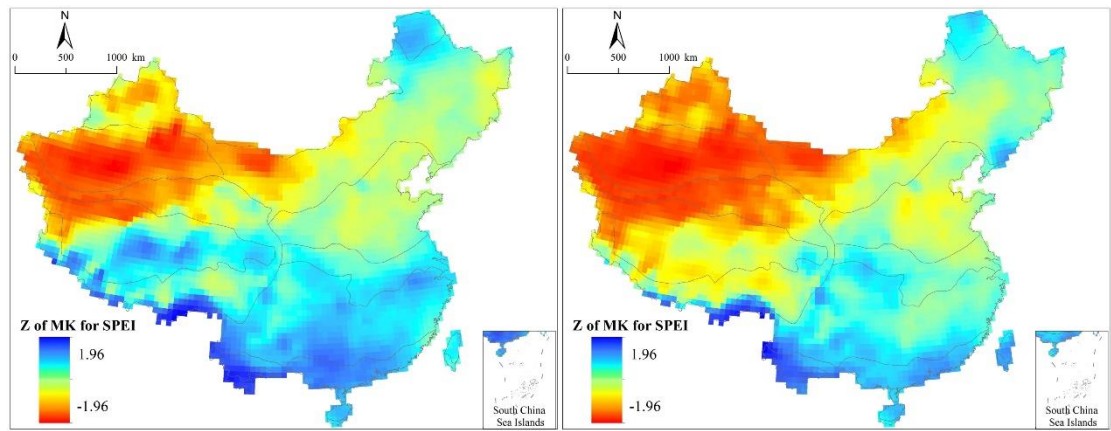





**Figure 4: SPEI M-K test results (Z) distributed across China for 2011–2099 in RCP4.5 (left) and RCP8.5 (right); Z values of M–K test (a < 0.05); Z > 1.96 represents a significant upward (wetting) trend and Z < -1.96 represents a significant downward (drying) trend.**

5   **Figure 5: Spatial distribution of moderate drought frequency (-2.0 < SPEI ≤ -1.0) in different time periods (1: 2011– 2040; 2: 2041–2070; 3: 2071–2099) under RCP4.5 (a) and RCP8.5 (b) scenarios.**



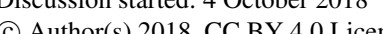



**Figure 6: Spatial distribution of extreme drought frequency (SPEI ≤ -2.0) in different time periods (1: 2011–2040; 2: 2041–2070; 3: 2071–2099) under RCP4.5 (a) and RCP8.5 (b) scenarios.**


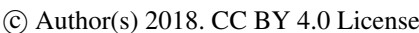


**Figure 7: Spatial distribution of drought duration in different time periods (1: 2011–2040; 2: 2041–2070; 3: 2071–2099) under RCP4.5 (a) and RCP8.5 (b) scenarios.**