# Peer review of "Spatial and Temporal Pattern of Drought Hazard under Different RCP Scenarios for China in the 21[st] century"

_Natural Hazards and Earth System Sciences, 2018_

## Referee Comment (RC1) · Anonymous Referee #1 · 6 Dec 2018

This research evaluated the future drought conditions of China by calculating drought index SPEI. The research is reasonable and the illustrations are comprehensive. My specific comments are listed below: 1. Since your research focuses on the SPEI in the future stage, the bias correction method you utilized to correct GCM data should be described in details. 2.The advantages of SPEI cannot be seen if you only employ one drought index. If you compare SPEI with the other one that does not take temperature into consideration (e.g. SPI), then we can clearly understand the importance of temperature or ET In drought monitoring. 3.The item "ET0" usually refers to reference evapotranspiration. Here in the calculation of SPEI, ETp or PET are recommended.

---

## Referee Comment (RC2) · He (Referee) · 7 Dec 2018

This manuscript presents an important study on the spatial and temporal pattern of drought hazard under different RCP scenarios for China which is a hot topic in climate change and disaster risk researches. The results gave clear maps of the disaster hazards under RCP 4.5 and 8.5 respectively for 2011–2099 over China. The conclusions are very useful for risk assessment and management. Overall, this paper is well conducted and organized. The method and data were suitable and reliable. It presents a detailed and robust analysis of the spatial and temporal variation in drought hazards in three time periods. The conclusions provide an important reference for adapting to

extreme climate change and the prevention and reduction of disaster risks. For these reasons, I would like to recommend publish this paper on NHESS after some minor revisions. The detailed comments are as follows.

Page1, line 13: "...to evaluate drought" should be "...to evaluate drought hazard" Page1, line 14: "(RCP) 4.5 and 8.5 climate data" should be "(RCP) 4.5 and 8.5 scenarios climate data" Page1, line 29: change "are" to "were" Page1, line 31: suggest add "for disasters" after "Therefore, risk assessment" Page1, line 32: "Risk is often represented..." could be "Risk of disaster is often represented..." Page2, line 3: it's better to revise "Evaluating hazards is the most important aspect..." to "Evaluating hazards for disasters is the most important aspect..." Page2, line 3: "extensive research has been conducted..."should be "extensive researches have been conducted..." Page2, line 7: "...in drought in semi-arid areas in the western United States...", too many same prepositions "in" in one sentence Page2, line 8: "...the drought situation", here the word "situation" is not very good Page2, line 10: "Similar research", it's better to use plural Page2, line 11: is "Xing-Guo" a family name? Page2, line 13: You could use "In addition,..." instead of "Finally, ..." Page 2, line 23: "Drought indexes..." could be "The main drought indexes..." Page 2, line 24: add "et al." after "...Rhee et al., 2010)" Page 2, line 31: "The primary objective of this study was to..." should be "The primary objective of this study is to..." Page 3: for section 2.1, you'd better to use 1-2 sentences to describe what is "Eco-geographical regionalization" Page 3, line 11: "included" should be "includes" Page 4, line 14: "indices", you use "indexes" in page2 line 22, should be consistent throughout the context Page5, line2: the title of section 3.1 "Determining SPEI over different timescales", what do you want to determine in this section, SPEI or time scale? Page5, line23-24: rewrite this sentence to make it more clear Page 6, line 9: This sentence is a little bit redundant Page 7 line 18: change "continuing" to "continuous"
* * *

---

## Author Comment (AC1) · 14 Dec 2018

Dear Editors and Reviewers: Thank you for your letter and for the reviewer's comments concerning our manuscript entitled "Spatial and Temporal Pattern of Drought Hazard under Different RCP Scenarios for China in the 21st century" (ID: NHESS-2018-242). Those comments are all valuable and very helpful for revising and improving our manuscript. We studied comments carefully and made corrections in the manuscript. The response to the reviewer's comments are as follow:

1. Since your research focuses on the SPEI in the future stage, the bias correction method you utilized to correct GCM data should be described in details.

[Figure]

Authors' response: Thanks for your suggestion. We have supplemented bias correction method to correct GCM data obtained from ISI-MIP in Section 2.2. The statement is "Statistical bias correction methods facilitate the comparison of observed and simulated impacts during the historical reference period and a continuous transition into the future (Rötter et al., 2011). The applied method for correcting simulated data of ISI-MIP is that preserves the absolute changes in monthly temperature, and relative changes in monthly values of precipitation and the other variables (Hempel et al., 2013). For different climate variables, temperature is corrected by additive correction while precipitation, wind speed, relative humidity and solar radiation is corrected by multiplicative correction". Besides, the references were added, too.

References:

Hempel, S., Frieler, K., Warszawski, L., Schewe, J., and Piontek, F.: A trend-preserving bias correction &ndash&59; the ISI-MIP approach, Earth System Dynamics,4,2(2013-07-31), 4, 219-236, 2013.

Rötter, R. P., Carter, T. R., Olesen, J. E., and Porter, J. R.: Crop-climate models need an overhaul, Nature Climate Change, 1, 175-177, 2011.

2. The advantages of SPEI cannot be seen if you only employ one drought index. If you compare SPEI with the other one that does not take temperature into consideration (e.g. SPI), then we can clearly understand the importance of temperature or ET in drought monitoring.

Authors' response: Thanks for your comment. As we stated in the 3rd paragraph of Section 1, we chose SPEI as the drought index to analyze in our study after comparing the characteristics of the three most widely used indexes, SPI, PDSI and SPEI. SPEI combines the multi-scale characteristics of SPI and sensitive to warming characteristics of PDSI, which is so suitable to estimate drought hazard under climate change. But we also think it is really important to compare the results calculated by other drought indexes (e.g. SPI) as reviewer suggested, so we have supplemented the statement

in Section 4. The expression is "This general result is in agreement with those from Huang et al. (2017) and also proves the applicability of SPEI as a drought index in the context of climate change as previous studies. For example, Tan et al (2015) assessed drought hazard in Ningxia province of China indicated by SPI and SPEI, results showed that drying trends revealed by the SPEI were more significant than the SPI, and the trend magnitude was much greater. The differences between this two indexes are mainly attributed to the variation of temperature. It is confirmed that higher atmospheric evaporative demand resulting from temperature rise increased the severity of droughts (Vicenteserrano et al., 2014). Therefore, the SPEI that considers both precipitation and evapotranspiration is more suitable than the SPI for applications examining drought hazard under climate change." Besides, the references were added, too.

References:

Tan, C., Yang, J., and Li, M.: Temporal-Spatial Variation of Drought Indicated by SPI and SPEI in Ningxia Hui Autonomous Region, China, Atmosphere, 6, 1399-1421, 2015.

Vicenteserrano, S. M., Lopezmoreno, J., Beguería, S., Lorenzolacruz, J., Sanchezlorenzo, A., Garcíaruiz, J. M., Azorinmolina, C., Morántejeda, E., Revuelto, J., and Trigo, R.: Evidence of increasing drought severity caused by temperature rise in southern Europe, Environmental Research Letters, 9, 044001, 2014.

3. The item "ET0" usually refers to reference evapotranspiration. Here in the calculation of SPEI, ETp or PET are recommended

Authors' response: Thanks for your advice. After checking related literatures, Potential evapotranspiration (PET) is defined as the amount of evaporation that would occur if a sufficient water source was available. Often a value for the potential evapotranspiration is calculated at a nearby climatic station on a reference surface, conventionally short grass. This value is called the reference evapotranspiration (ET0). Since the literature proposed SPEI described as D=P-PET (Vicenteserrano et al., 2010), we have modified the expression to "PET" and make it to be consistent with the original text in Section

2.3. Also, we have supplemented the definition of PET in Section 2.3 to make it clearer to understand.

Reference:

Vicenteserrano, S. M., Beguería, S., and Lópezmoreno, J. I.: A multiscalar drought index sensitive to global warming: the standardized precipitation evapotranspiration index, J. Clim., 23, 1696-1718, 2010.

---

## Author Comment (AC2) · 14 Dec 2018

Dear Editors and Reviewers: Thank you for your letter and for the reviewer's comments concerning our manuscript entitled "Spatial and Temporal Pattern of Drought Hazard under Different RCP Scenarios for China in the 21st century" (ID: NHESS-2018-242). Those comments are all valuable and very helpful for revising and improving our manuscript. We studied comments carefully and made corrections in the manuscript. The response to the reviewer's comments are as follow:

1. Page1, line 13: ". . .to evaluate drought" should be ". . .to evaluate drought hazard"

[Figure]

Authors' response: Thanks for your suggestion. The statement was rephrased to "...to evaluate drought hazard".

2. Page1, line 14: "(RCP) 4.5 and 8.5 climate data" should be "(RCP) 4.5 and 8.5 scenarios climate data"

Authors' response: Thanks for your advice. The expression was rephrased to "(RCP) 4.5 and 8.5 scenarios climate data".

3. Page1, line 29: change "are" to "were"

Authors' response: Thanks for your suggestion. The statement was rephrased to "were".

4. Page1, line 31: suggest add "for disasters" after "Therefore, risk assessment"

Authors' response: Thanks for your suggestions. The statement was rephrased to "Therefore, risk assessment for disasters..."

5. Page1, line 32: "Risk is often represented..." could be "Risk of disaster is often represented..."

Authors' response: Thanks for your advice. The expression was rephrased to "Risk of disaster is often represented..."

6. Page2, line 3: it's better to revise "Evaluating hazards is the most important aspect..." to "Evaluating hazards for disasters is the most important aspect..."

Authors' response: Thanks for your advice. The statement was modified to "Evaluating hazards for disasters is the most important aspect..."

7. Page2, line 5: "extensive research has been conducted..."should be "extensive researches have been conducted..."

Authors' response: Thanks for your suggestions. The statement was rephrased to "extensive researches have been conducted..."

8. Page2, line 7: "...in drought in semi-arid areas in the western United States...", too many same prepositions "in" in one sentence

Authors' response: Thanks for your advice. The expression was rephrased to "Gutzler and Robbins (2011) focused on drought changes in semi-arid areas of the western United States based on the Palmer Drought Severity Index (PDSI)..."

9. Page2, line 8: "...the drought situation", here the word "situation" is not very good

Authors' response: Thanks for your suggestion. The word "situation" was modified to "hazard".

10. Page2, line 10: "Similar research", it's better to use plural

Authors' response: Thanks for your suggestion. The statement was rephrased to "Similar researches in China have..."

11. Page2, line 11: is "Xing-Guo" a family name?

Authors' response: Thanks for your comment. Sorry for our incorrect citation. We have modified it to "Hu et al., 2015" and updated the citation in Section References.

Reference

Hu, S., Mo, S., and Lin, Z.: Projections of spatial-temporal variation of drought in north China, Arid. Land. Geogr., 2015.

12. Page2, line 13: You could use "In addition,..." instead of "Finally, ..."

Authors' response: Thanks for your suggestion. The statement was rephrased to "In addition,..."

13. Page 2, line 23: "Drought indexes..." could be "The main drought indexes..."

Authors' response: Thanks for your advice. The expression was modified to "The main drought indexes..."

[Figure]

14. Page 2, line 24: add "et al." after "...Rhee et al., 2010)"

Authors' response: Thanks for your advice. "et al." was added after "...Rhee et al., 2010)".

15. Page 2, line 31: "The primary objective of this study was to..." should be "The primary objective of this study is to..."

Authors' response: Thanks for your suggestion. The statement was rephrased to "The primary objective of this study is to..."

16. Page 3: for section 2.1, you'd better to use 1-2 sentences to describe what is "Eco-geographical regionalization"

Authors' response: Thanks for your advice. We have supplemented what is "Eco-geographical regionalization" and rewrote contents in Section 2.1 to make it clearer and easier to understand. The expression is "Eco-geographical regionalization for China adopts the principle of adapting to the laws of natural geographical differentiation, and divides the country into 11 temperature zones according to differences in surface topographic features and geomorphic structure, combinations of temperature and water conditions, and zonal vegetation and soil type (Fig. 1) (Zheng, 2008). To reveal regional differences in drought hazard, eco-geographical regionalization for China was used for analysis in this study. The equatorial tropical zone was not included for lacking in data."

17. Page 3, line 11: "included" should be "includes"

Authors' response: Thanks for your suggestion. The statement was rephrased to "includes".

18. Page 4, line 14: "indices", you use "indexes" in page2 line 22, should be consistent throughout the context

Authors' response: Thanks for your advice. We have modified "indices" to "indexes" in

Page 4, line 14, Page 4, line 15 and Page 2, line 20 to make the statement consistent throughout the context.

19. Page5, line2: the title of section 3.1 "Determining SPEI over different timescales", what do you want to determine in this section, SPEI or time scale?

Authors' response: Thanks for your comment. Sorry for our inappropriate expression. The main objective of Section 3.1 is determining the most suitable timescale of SPEI for the M-K test, and to calculate the frequencies and durations of droughts. We have changed the title of Section 3.1 to "Determining timescale of SPEI for analysis" to make it clearer to understand.

20. Page5, line23-24: rewrite this sentence to make it more clear

Authors' response: Thanks for your advice. This sentence was rewritten to "In RCP4.5, most of north China shows downward (drying) trends, among which middle and western of MT, western of WT and northwest of PT present significant downward trends at the 0.05 significance level."

21. Page 6, line 9: This sentence is a little bit redundant

Authors' response: Thanks for your suggestion. We have deleted this sentence and added "As for moderate drought frequency (Fig 5)" at the beginning of this paragraph.

22. Page 7 line 18: change "continuing" to "continuous"

Authors' response: Thanks for your suggestions. The statement was change to "continuous".